# High Initial Dose of Monitored Vitamin D Supplementation in Preterm Infants (HIDVID Trial): Study Protocol for a Randomized Controlled Study

**DOI:** 10.3390/nu16050700

**Published:** 2024-02-29

**Authors:** Dominika Paw, Renata Bokiniec, Alicja Kołodziejczyk-Nowotarska

**Affiliations:** Department of Neonatology and Neonatal Intensive Care, Medical University of Warsaw, 00-315 Warsaw, Poland; dominika.paw@wum.edu.pl (D.P.); renata.bokiniec@wum.edu.pl (R.B.)

**Keywords:** vitamin D, infant, premature, osteopenia, sepsis, interleukin-6, parathyroid hormone

## Abstract

Vitamin D deficiency can escalate prematurity bone disease in preterm infants and negatively influence their immature immunology system. Infants born at 24 + 0/7 weeks to 32 + 6/7 weeks of gestation will be considered for inclusion. Cord or vein blood samples will be obtained within 48 h after birth for 25-hydroxyvitamin D level measurements. Parathyroid hormone and interleukin-6 levels will be measured. Infants will be randomized to the monitored group (i.e., an initial dose of 1000 IU/day and possible modification) or the controlled group (i.e., 250 IU/day or 500 IU/day dose, depending on weight). Supplementation will be monitored up to a postconceptional age of 35 weeks. The primary endpoint is the percentage of infants with deficient or suboptimal 25-hydroxyvitamin D levels at 28 ± 2 days of age. 25-Hydroxyvitamin D levels will be measured at postconceptional age 35 ± 2 weeks. Secondary goals encompass assessing the occurrence of sepsis, osteopenia, hyperparathyroidism, and interleukin-6 concentration. The aim of this study is to evaluate the efficacy of monitored vitamin D supplementation in a group of preterm infants and ascertain if a high initial dosage of monitored vitamin D supplementation can decrease the occurrence of neonatal sepsis and metabolic bone disease.

## 1. Introduction

The well-established significance of vitamin D (vit D) in preserving skeletal health is widely recognized [1,2]. The identification of vit D receptors in various tissues and cells has offered fresh perspective on its biological effects [2]. Maintaining calcium and phosphate homeostasis is one of the key tasks for the proper functioning of the fetus and infant organism. Serum calcium and phosphorus levels are under hormonal control of the interdependent parathyroid hormone (PTH; secreted by the parathyroid glands), vit D (the active metabolite 1,25-dihydroxycholecalciferol is formed in the kidneys), phosphate (secreted by the bones), and, to a lesser extent, calcitonin (produced by the parafollicular thyroid) [3]. Parathormone (PTH)-like peptides, PTH, and calcitriol have a basic regulatory role in active transplacental calcium transport. After birth, the calcium concentration decreases in the blood; therefore, the PTH concentration increases, which stimulates the reabsorption of calcium in the kidneys and inhibits the reabsorption of phosphorus [3,4]. Furthermore, the increased synthesis of calcitriol promotes the active absorption of calcium and phosphorus from the intestines. However, in preterm infants, intestinal absorption is compromised, and human milk proves inadequate as a source of calcium and phosphorus during this developmental stage [5,6].

Research has indicated that neonatal vit D storage at birth, whether it be a full-term or preterm infant, depends on the maternal 25-hydroxyvitamin D [25(OH)D] concentration because a developing fetus obtains all of its vit D from the mother [7,8]. Deficient levels of vit D in pregnant women are linked to inadequate levels in their premature newborns (with a strong correlation, *p* value < 0.01) [9]. Vit D holds importance during pregnancy and lactation. Premature neonates miss the crucial phase of maximum mineral accretion occurring in the final trimester of pregnancy, during which 60% and 80% of whole-body calcium and phosphorus, respectively, are actively transferred from the mother to the fetus [10]. This coincides with a significant exponential rise in bone formation between 24 weeks’ gestation and term. Inadequate postnatal Ca, P, and vit D supplementation can lead to metabolic bone disease (MBD) of prematurity. MBD increases the risk of death by three times and extends hospitalization time by 40% [11]. Zung et al. conducted a prospective study in 2020, which showed that only 24% of infants had sufficient levels of 25(OH)D after birth. They suggested supplementation with at least 600 IU/day of vit D for premature newborns and infants with low birth weight to achieve increased levels of 25(OH)D before discharge [12].

Scientific societies unanimously recommend vit D supplementation in neonatal patients, as well as in pregnant women. The guidelines from the American Academy of Pediatrics (2013) suggest supplementation of 200–400 IU/day (depending on the weight) for enterally fed preterm infants [13]. In line with the latest recommendations from the European Society for Paediatric Gastroenterology, Hepatology, and Nutrition, vit D supplementation for preterm infants should range from 400 to 700 IU/kg/day, with a maximum daily dose capped at 1000 IU/day [14]. The argument for a higher dosage is the prevalence of a commonly occurring deficiency in pregnant women. In addition, the effect on the pleiotropic action of vit D in infants is observed at a concentration of 25(OH)D >30 ng/mL [7,14]. According to Polish guidelines, for neonates born at <33 weeks of gestation, if enteral nutrition is feasible, a dosage of 800 IU/day (20 µg/day) of cholecalciferol is recommended during the initial month of life, irrespective of the feeding method [7]. Dietary intake calculations should commence in the second month of life. Monitoring of supplementation is advised through serum 25(OH)D concentration assays, conducted during hospitalization (with the initial assessment after 4 weeks of supplementation) and/or follow-up in outpatient care. Caution is advised as a total daily cholecalciferol dose >1000 IU (i.e., 25 µg/day) may pose a risk of vit D overdose, particularly in neonates with a birth weight <1000 g.

A previous study [15] conducted in the Department of Neonatology and Neonatal Intensive Care at the Medical University of Warsaw (Warsaw, Poland) showed that approximately 45% of patients had suboptimal and deficiency levels at 4 weeks of age on the initial supplementation of 500 IU. Infants were randomized to receive monitored supplementation (with the option of dose modification, based on 25(OH)D levels, as per protocol) or to receive standard therapy (i.e., 500 IU) for up to 52 weeks of postconceptional age (PCA). However, monitoring has been proven to avoid overdosing during further follow-up. The quantitative ultrasonography assessment of average bone mass did not differ between the two groups of patients, but the initial dose was the same in both groups.

Vit D also has an important role in immunoregulation in the pediatric population [16]. In contrast to the immune system of full-term infants, the immune system of preterm infants exhibits several notable differences. These include a reduced population of monocytes and neutrophils, compromised effectiveness of these cells in eliminating pathogens, and diminished cytokine production. Consequently, there is a restricted capacity for T cell activation, leading to a decreased ability to combat bacterial infections and identify viruses within cells [14,15]. These abnormalities can contribute to the development of late-onset sepsis, a prevalent and serious condition in neonatal intensive care units. The frequency of late-onset sepsis is linked to the level of prematurity, exhibiting geographical variations ranging from 0.61% to 14.2% among hospitalized newborns [17]. The incidence of sepsis is significantly higher (8–20%) in premature infants and in infants with low birth weight (i.e., <1000 g) [18]. The rapid progression of the disease may lead to secondary organ damage. Early treatment is needed, which is very difficult and challenging to achieve in clinical practice. The overall mortality rate of late-onset sepsis is 2% to 20% [18]. The aim of this study will be to assess the effectiveness of monitored vit D supplementation in a population of preterm infants and to identify whether proper vit D supplementation in preterm infants can reduce the incidence of neonatal sepsis and metabolic bone disease.

## 2. Materials and Methods

### 2.1. Study Design

We are implementing an unblinded, parallel-group, randomized controlled superiority trial. Parents of neonates born between 24 + 0/7 weeks and 32 + 6/7 weeks of gestation will be offered the opportunity to participate in the study within the first 24 h of the infant’s life. After providing information orally and in writing concerning the study, we will seek written consent from the parents for the neonate and mother to participate in the trial. To enhance the relevance of the research findings, we will stratify the study group by gestational age into 24–28 and 29–32 weeks of gestation subgroups. We have planned to recruit for the study for 36 months, from September 2024 to the end of 2027. The study protocol has been registered at ClinicalTrials.gov (identifier: NCT06199102). Please see Table 1 for all planned patient-related activities.

### 2.2. Setting and Participants

All infants admitted to the Department of Neonatology and Intensive Care at the Medical University of Warsaw (Warsaw, Poland) and born at 24 + 0/7 weeks to 32 + 6/7 weeks of gestation will be considered for inclusion. Our tertiary perinatal center has an annual number of births of approximately 2500 infants, including an average of 80 infants aged <33 weeks of gestation. 25(OHD) neonatal birth concentration strongly correlates with pregnant vit D supplementation. According to the Polish Society of Gynecologists and Obstetricians, it is recommended to supplement vit D at a dose of 1500–2000 IU/day during pregnancy [19]. However, the available multivitamin supplements for pregnant women in Poland vary from 400 IU to 2000 IU. The parents of the infants will provide informed consent. Patients will be randomly assigned to the monitored group or the controlled group. We will collect a cord or vein blood sample within the first 48 h of life to evaluate 25(OH)D levels at birth. Blood samples will be taken simultaneously with other laboratory tests so as not to expose the newborn to additional blood donations. Additionally, the maternal 25(OH)D level and the neonatal birth levels of PTH and IL-6 will be measured. Once infants reach a daily enteral feeding volume of 30 mL/kg, they will receive an initial vitamin D dose. We will continue to monitor the supplementation until 35 weeks of PCA.

### 2.3. Inclusion Criteria

We will include all preterm infants with a gestational age of 24 + 0/7 to 32 + 6/7 born at our clinic or admitted to the intensive care unit (outborn infants need to be admitted within 48 h after delivery). Upon recruitment, caregivers are required to provide written informed consent for both the mother and her child to participate in the study.

### 2.4. Exclusion Criteria

We will exclude infants born at >32 weeks of gestation and with major congenital abnormalities or other severe congenital malformations, infants with genetic disorders (diagnosed before and after birth) deemed incompatible with survival, and infants with diagnosed cholestasis. Cholestasis is defined as direct bilirubin > 1 mg/dL (>17.1 μmol/L) [20]. Exclusion criteria also include the lack of written informed consent and communication challenges with caregivers.

### 2.5. Randomization Criteria

Following the receipt of signed consent for participation from the parents of the neonates, the participants will be registered and randomized within the initial 48 h. Enrolled infants will be randomized (1:1 allocation) with a block size of six.

### 2.6. Intervention

Enrollment will occur within 48 h from birth, typically within the first 24 h of life. During this process, patients will undergo screening for both inclusion and exclusion criteria.

Infants in the monitored group will receive an initial dose of 1000 IU of vit D (cholecalciferol/Devikap; Polpharma, Starogard Gdański, Poland), whereas infants in the controlled group will receive 250 IU for extremely low birth weight infants and 500 IU for infants weighing above 1000 g. The administration of vit D will occur just before feeding via an orogastric tube, until the infants commence bottle feeding directly by mouth. Parenteral nutrition includes an additional 160 IU/kg of vit D, and enteral nutrition comprises 150–300 IU/kg, depending on the quantity and source of enteral feeding (e.g., human milk fortifiers or milk formula). At 28 ± 2 days of age, blood samples will be obtained for 25(OH)D concentration measurement, followed by measurements every 4 weeks and/or 35 ± 1 weeks of PCA. In the monitored group, vit D doses will be adjusted accordingly based on the 25(OH)D levels, following the scheme outlined in the Polish recommendation (Table 2) [7]. Dietary intake will be calculated starting in the 2nd month of life.

Infants assigned to the standard therapy group will undergo the same blood sample collection procedure as the monitored group, but without any alterations in their dosing regimen. If we observe a potentially toxic level of >100 ng/mL, we will stop supplementation in both groups for ethical reasons.

### 2.7. Primary Outcome

The percentage of infants with deficient or suboptimal 25(OH)D levels at 28 ± 2 days of age will be the primary endpoint. The 25(OH)D levels will additionally be measured at 35 ± 1 weeks of PCA. Neonatal staff will obtain venous samples of 0.3 mL to assess the 25(OH)D serum concentration. The study site will utilize the VIDAS automated quantitative test (bioMérieux, Marcy l’Etoile, France) to measure 25(OH)D concentrations. In Poland, the diagnostic thresholds for serum 25(OH)D concentrations are as follows [7]: <20 ng/mL (50 nmol/L) indicates vitamin D deficiency;
>20 ng/mL (50 nmol/L) and <30 ng/mL (75 nmol/L);≥30 ng/mL (75 nmol/L) up to 50 ng/mL (125 nmol/L) reflect adequate to optimal vitamin D status;>50 ng/mL (125 nmol/L) up to 100 ng/mL (250 nmol/L) indicate a high vitamin D supply;>100 ng/mL (250 nmol/L) reflect an increased risk for intoxication [7].

### 2.8. Secondary Outcomes

#### 2.8.1. Sepsis

Neonatal sepsis, a frequently encountered clinical syndrome in newborns, is associated with elevated morbidity and mortality rates. Current research highlights the major regulatory role of vit D in the immune system, which is capable of impeding cell proliferation and maturation while also controlling inflammatory cytokines [21]. Sepsis is defined as a clinical condition, the cause of which is a systemic inflammatory reaction of the body arising because of bacterial infection, primarily under the influence of released bacterial toxins and fragments constituting bacterial components. Research findings indicated a significant association between neonatal sepsis and low levels of vit D in both cord blood and maternal blood [22]. Late-onset sepsis is defined as blood culture-proven and/or clinical sepsis occurring after 3 days of age [18]. Per the guidelines of the Polish Society of Neonatology, the initial suspicion of sepsis is based on the presence of at least two of the following clinical symptoms: apnea, elevated body temperature, signs of respiratory distress syndrome, hypoxia, feeding reluctance, somnolence, or overstimulation. Positive laboratory test results further support the suspicion. The confirmation of a diagnosis is a positive bacteriological blood test result from one blood sample of at least 1 mL [23]. Neonatal staff will collect venous samples (1 mL) for a bacteriological blood test and laboratory tests. Laboratory tests include morphology with smear; if any sepsis risk factor exists, the findings will have an immature-to-total neutrophil ratio (I:T) >0.2 and levels of C-reactive protein (CRP) >10 mg/L, procalcitonin (PCT) >0.5 ng/mL, and interleukin 6 (IL-6) >44 pg/mL [23,24].

#### 2.8.2. Bone Mass and Biochemical Markers of MBD

Osteopenia in preterm infants occurs in approximately 20–30% of children with a birth weight < 1500 g and in 50–60% of children with a birth weight < 1000 g [11]. We have opted to define MBD as a reduction in bone mineral content compared to the anticipated level of mineralization for a fetus or infant of similar size or gestational age. This definition is coupled with observed biochemical and/or ultrasound changes. Neonatal staff will obtain venous samples of 0.7 mL for the Ca-P metabolism tests, including serum alkaline phosphatase and phosphate levels at 35 ± 1 weeks of PCA. The measurements will be performed using the AU480 chemistry analyzer (Beckman Coulter, Brea, CA, USA), which will undergo calibration checks with suitable controls following product guidelines. The mean value of these measurements will be used for data analysis.

We decided to define MBD as serum levels of alkaline phosphatase > 500 IU and serum phosphate < 1.8 mmol/L [25]. Moreover, we intend to evaluate average bone mass through the use of quantitative ultrasound (Sunlight PREMIER 7000; BeamMed, Petah Tikva, Israel). This method, deemed safe, noninvasive, radiation-free, and user-friendly, has been proposed as a screening tool for detecting osteopenia in premature infants [25,26]. Using a compact ultrasound probe (CRB probe RoHS 900–1000 kHz) positioned along the mid-tibia, this device quantifies the speed of sound (m/s) through the axial transmission mode. However, due to significant intraindividual variation, establishing normal values becomes challenging. The measurements will be made on the tibia, with the mid-tibial shaft length determined by measuring the distance from the knee to the heel. Placing the probe over the medial aspect of the mid-shaft tibia, three speed-of-sound measurements will be conducted.

#### 2.8.3. Parathormone

PTH is a peptide hormone that undergoes upregulation in response to hypocalcemia and vit D deficiency. It stimulates the reabsorption of calcium in the renal tubules, inhibits the excretion of phosphates, stimulates osteolysis of osteocytes and bone resorption by osteoclasts, and increases the absorption of calcium from the intestines [4]. Hyperparathyroidism may further worsen rickets through the promotion of osteolysis. Serum or plasma concentration of PTH in infants should be 10–40 pg/mL. Venous samples (0.3 mL) will be collected by neonatal staff for serum PTH concentration measurement at birth, at 28 ± 2 days of life, and at 35 ± 1 weeks of PCA, using the CMIA method by Abbot Alinity Analyzer. The calibration of the analyzer will be verified using suitable controls, in accordance with the product guidelines.

#### 2.8.4. Interleukin-6

IL-6 is a reliable marker for predicting neonatal sepsis [27,28]. It is released within 2 h after the onset of bacteremia, reaching its peak at approximately 6 h and subsequently declining over the next 24 h. The accuracy of IL-6 as a predictor is higher in preterm infants compared to that in mixed-study populations. Based on the study by Prinsen et al., the reference interval is calculated as 44 pg/mL [29]. The established cutoff can now be utilized to measure IL-6 as an early pro-inflammatory marker in neonates. Three drops of capillary blood will be collected by neonatal staff for IL-6 levels at birth, at 28 ± 2 days of life, and at 35 ± 1 weeks of PCA, and in any case of suspected neonatal late-onset sepsis. The analyzer’s calibration will be checked with appropriate controls, per product guidelines (POCT model VMFIA1001; BISAF, Wrocław, Poland).

### 2.9. Adverse Events

We will define an adverse event as any unexpected medical incident in an infant, irrespective of a potential causal connection. Every adverse event that takes place from the time of entry into the study until hospital discharge will be documented and recorded. If an adverse event meets the criteria for a serious adverse event (SAE) during this period, it will be reported to the local ethics committee. In the context of this study, a SAE is characterized as any unforeseen medical incident deemed by the investigators to be directly associated with the study intervention, leading to a life-threatening condition, severe or permanent disability, or prolonged hospitalization. SAEs that occur after a participant is removed from the study will not be reported unless investigators suspect a correlation with the study drug or a procedure outlined in the study protocol. The adverse effects of hypervitaminosis of vit D described in the literature are nephrocalcinosis and nephrolithiasis [30,31].

#### 2.9.1. Nephrocalcinosis and Nephrolithiasis

Neonatal staff will collect venous samples (0.7 mL) for the Ca-P metabolism tests, including serum and urine calcium, and creatinine level measurements at 28 ± 2 days of life and at 35 ±1 weeks of PCA. The measurements will be performed using the AU480 chemistry analyzer from Beckman Coulter. The calibration of the analyzer will be verified with suitable controls, following the product guidelines. Hypercalcemia will be defined as a serum level of ≥2.75 mmol/L. Hypercalciuria will be assessed by determining urine calcium-to-creatinine ratios, which vary, depending on the individual’s age [32]. Malone et al. reported that infants born prematurely with lower birth weights and immature kidneys face heightened susceptibility to nephrocalcinosis, the risk factor of smaller kidneys, obstructive hydronephrosis, and hypertension [33]. They observed a trend toward higher average serum 25(OH)D concentrations among preterm infants with nephocalcinosis. The incidence of nephrocalcinosis was high at 41% (23/56) in infants born at ≤32 weeks gestational age. Therefore, monitoring of both the 25(OH)D level and serum and urine calcium levels is recommended in this group of premature neonates. In infants, excess levels of urine calcium and creatinine are considered risk factors for nephrolithiasis. At 35 ± 1 weeks of PCA, infants will undergo a nephrocalcinosis assessment conducted by two independent and skilled ultrasonographers (blinded to the group and other ultrasonographers) using the HD11 XE ultrasound system (Philips Healthcare, Andover, MA, USA). Nephrocalcinosis will be identified by heightened medullary echogenicity, characterized by small white flecks at the tip of the pyramids. Photographic documentation will be captured.

#### 2.9.2. Vit D Intoxication

Vit D intoxication is characterized by a serum 25(OH)D level exceeding 100 ng/mL, leading to hypercalcemia, hypercalciuria, and the suppression of PTH [7]. In cases of vit D overdose treatment, ceasing therapy immediately is crucial. Therefore, calcium levels and urinary calcium excretion should be monitored while also tracking serum 25(OH)D concentrations at 1-month intervals until they reach a level below or equal to 50 ng/mL. Once normocalcemia, normocalciuria, and 25(OH)D concentrations below or equal to 50 ng/mL are achieved, resuming prophylactic or therapeutic interventions can be considered, provided that vit D hypersensitivity has been ruled out.

### 2.10. Retention of Participants in the Study

Given that a significant portion of the enrolled patients require long-term hospitalization, our primary focus will be on effective staff training and ensuring the trial is well-managed by the study team. To address any concerns arising during the trial, we will organize monthly departmental meetings.

### 2.11. Data Monitoring

We opted not to establish a data monitoring committee since the intervention in the trial (i.e., vit D supplementation of 250–1000 IU/day) aligns with the standard of care supported by multiple pediatric societies [7,13,14]. Moreover, we are well-informed about the potential adverse effects associated with this intervention.

### 2.12. Sample Size Calculations

The sample size was determined based on the primary outcome, which is defined as the count of neonates with 25(OH)D deficiency or insufficiency. A previous study conducted in our department reported an incidence of vit D deficiency or vit D insufficiency of 80% in preterm infants at birth, which was reduced with vit D supplementation to 45% at 28 ± 2 days of age. Assuming an estimated prevalence of 80% in the standard supplementation group, we chose to detect a decrease of 25% in patients with vitamin D deficiency and vitamin D insufficiency versus the control group with a power of 80% and α = 0.05 to meet acceptable recruitment rates and reach statistically significant results. Hence, 54 infants are required in each study group. Assuming a 20% loss to follow-up (due to the decision of the principal investigator, withdrawal of consent from a parent/legal guardian), the sample size has been set at 130 patients and their mothers. We believe the recruitment of patients will last approximately 3 years.

### 2.13. Statistical Analysis

Statistical analysis will be conducted using Statistica 13.1 (StatSoft, Tulsa, OK, USA). Our intention is to conduct intention-to-treat analyses. Continuous data will be presented as means and standard deviations or as medians and ranges, while categorical variables will be expressed as proportions. Normally distributed continuous variables will undergo analysis using the Student’s *t*-test, while the Wilcoxon rank-sum test will be applied for skewed data. Categorical variables will be analyzed using either the chi-square test or Fisher’s exact test. The 95% confidence intervals (95% CIs) will be calculated for relative risk, as well as risk differences for categorical variables, and mean differences with 95% CIs will be calculated for continuous variables. We plan to analyze the partial data once annually during the recruitment of patients.

## 3. Discussion

This study aims to evaluate the effectiveness of monitored vit D supplementation in a population of preterm infants and determine whether appropriate vit D supplementation can reduce the incidence of neonatal sepsis and MBD. A meta-analysis published in 2022 on pediatric enteral vit D supplementation reported improved growth and vit D status in preterm and low-birth-weight infants [34]. However, the benefits of a higher dose of supplementation were not found in serious neonatal morbidities except by very low-to-low certainty evidence of a reduction in serum PTH and vit D levels in high-dose vit D supplemented groups. The meta-analysis included 14 trials comprising high versus low supplementation; however, the levels of heterogeneity and risk of bias were high. There is no consensus on the optimal dosage of vit D supplementation for preterm infants. A crucial factor to note is that preterm infants, in particular, are highly vulnerable to the adverse effects of vit D deficiency, given that 80% of the transfer of calcium and phosphorus from the placenta occurs between 24 weeks and 40 weeks of gestation [7,8,9].

There is also controversy regarding the methodology for measuring vit D. Miller et al. suggest that neonatal 25(OH)D measurements alone should not be used for assessing nutritional status due to the unreliability of the method for measuring 25(OH)D in preterm infants. In their opinion, clinical correlation and other laboratory parameters, including ionized calcium, should be considered [35]. In our study, we have chosen to use an immunoenzymatic method with the biological potential limitations of an overestimation of vit D status due to cross-reactivity to epimer. Also, the upper limit of 25(OH)D concentration in preterm infants is controversial. However, to assess vit D toxicity, we will measure serum and urine calcium levels, to prevent nephrocalcinosis, according to Malone et al. [33]. Currently, there is limited research on the immunological regulatory mechanism of vit D in the context of neonatal sepsis.

## Figures and Tables

**Table 1 nutrients-16-00700-t001:** Schedule of enrollment, interventions, and assessments.

Timepoint	Study Phase				
	Enrollment	Allocation	Postallocation		Close-Out
	First 48 h of Life	First 48 h of Life	4 Weeks of Age	8 Weeks of Age *	35 ± 2 Weeks of PCA
Eligibility screen	X				
Participation offer	X				
Prerandomization questionnaire	X				
Informed consent form	X				
Inclusion and exclusion criteria	X				
Randomization		X			
**Intervention**					
	**First 48 h** **of Life**	**First 48 h of Life**	**4 Weeks of Age**	**8 Weeks of Age ***	**35 ± 2 Weeks** **of PCA**
Monitored group			X	X	X
Control group			X	X	X
**Assessment**					
	**First 48 h** **of Life**	**First 48 h** **of Life**	**4 Weeks of Age**	**8 Weeks of Age ***	**35 ± 2 Weeks** **of PCA**
Blood sample for 25(OH)D		X	X	X	X
Blood sample for PTH		X	X	X	X
Blood sample for IL-6		X	**	**	X
Sample collection for Ca/P metabolism			X	X	X
Renal ultrasound					X
Bone mass assessment					X
Adverse events		X ***	X ***	X ***	X ***

25(OH)D: 25-hydroxyvitamin D; PCA: postconceptional age; IL-6: interleukin 6; Ca/P: calcium/phosphorus; IL-6: interleukin-6. * Eight weeks postconceptional age, only for infants born before 26 weeks of gestation. ** Blood sample for IL-6 will be obtained; for a suspected case of late-onset sepsis, a bacteriological blood test will be conducted simultaneously, using a blood sample of at least 1 mL. *** Adverse events monitored from enrollment until hospital discharge.

**Table 2 nutrients-16-00700-t002:** Dosage change, depending on 25(OH)D serum concentration, based on Polish guidelines.

Rules of Vitamin D Supplementation						
Serum concentration of 25(OH)D	<20 ng/mL	20–30 ng/mL	30–50 ng/mL	50–75 ng/mL	>75 ng/mL	>100 ng/mL
Management	Increase supply by 100%	Increase supply by 50%	Maintain the current supply	Reduce the dose by 50%	No supplementation for 1–2 months, and then 50% of the previous dose	Stop supplementation, Assess calcemia and calcuria, consider resuming supplementation after reaching 25(OH)D concentration < 50 ng/mL with normocalcemia and normocalcuria

25(OH)D: 25-hydroxyvitamin D.

## Data Availability

The datasets generated and/or analyzed during the present study are available from the corresponding author on reasonable request.

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
