# Peer review of "High Initial Dose of Monitored Vitamin D Supplementation in Preterm Infants (HIDVID Trial): Study Protocol for a Randomized Controlled Study"

_nutrients, 2024, doi:10.3390/nu16050700_

Round 1
Reviewer 1 Report
Comments and Suggestions for Authors
The paper covers an interesting proposal of research, the results of which could be relevant to everyday clinical work.
However, the range of gestational age of the participants is very broad and thus huge differences in the subgroups (if analyzed) are expected. If the authors would stratify the whole group by subgroups according to GA (eg 24 - 26, 27 - 29, 30 - 32 weeks), the value of the research could have been greatly increased. However, the number of children involved should be significantly greater - perhaps this issue could be solved by multicentre study, with other Polish tertiary perinatal centers included. Such design could also shorten the time needed to recruit sufficient number of participants (predicition of > 3 years seems very long).
The authors wrongly define very low birth weight as a weight < 1000 g (see lines 88, 137-138); the correct definition is either extremely low birth weight or change to weight < 1500 g.
In 2.2 (Setting and participants) I am missing more information about the institution (such as tertiary perinatal center?, annual number of births with GS < 33 weeks?, inborn - outborn / prenatal transport?). Perhaps also including the information on the guidelines regarding maternal supplementation of vitamin D in pregnancy in Poland (if they exist - or a comment that there is no uniform recommendation regarding vit D supplementation in pregnancy).
As (according to my experience) most ELBW infants need at least 7-10 days to reach 40 mL/kg/day, this limit might mean very late introduction of vitamin D. Perhaps the authors could consider a lower threshold and thus earlier initial dose?
In 2.4 I miss criteria for the diagnosis of cholestasis.
In 2.6 some data on administration of vit D should be added - if oral, is it given directly to the mouth (as I assume from the Devikap product specification, 1 drop = 500 IU) or by gastric tube (directly or diluted in some milk?). As vitamin D binds to the plastic wall this could have a significant effect on the actual amount of the vitamin received by the patient.
In Table 1, I suggest the term Serum concentration of 25(OH)D - instead of vitamin D (in the 1st column). The 2ndcolumn number should probably be <20 ng/mL (or a column 10 - 12 ng/mL is missing?). There are mistakes in the units in the columns with 50-75 and >75 ng - should be mL, not dL.
The limit of >100 ng/mL (line 152) differs from the highest level in Table 1.
The 2.7 (Primary outcome) should be combined with Table 1 - as it is mostly repetition of the same data.
I find citing "neonatal nurses" at each blood collection redundant (lines 189, 224, 234, 254) as the type of sample (venous, capillary, arterial) is important, as well as the quantity of blood (which is not listed!) and not the profession of the person who withdraws it.
The diagnosis for nephrolithiasis (or better - nephrocalcinosis, as stated in line 264) made by "a trained ultrasonographer" might be subjective and biased - I advise that two independent (and blinded for the group and the result of the other ultrasonographer) assessors are included.
Table 2 is insufficiently clear and descriptive - particularly as Enrollment and Allocation should be done in the first 24 hours (see line 99), lines Intervention are even blank (or something is missing).
As there are new data on analytical unreability of 25-OH-D vitamin measurements in preterm neonates (see Miller JJ et al, J Appl Lab Med 2023) I advise at least mentioning this bias in the discussion - on the other hand, inclusion of the value of ionized calcium as another laboratory parameter in the methods would be advised.
Although an extensive list of relevant literature is provided, I miss some recent citations, such as Jamali Z et al, BMC Pediatr 2023; Kumar M et al, Pediatrics 2022; Malone Jenkins S et al, Pediatr Nephrol 2022; Zung A et al, J Pediatr Endocrinol Metabol 2022.
Reference 16 is incomplete/false - could not be found in PubMed.
Author Response
Thank you for your review of our paper. We have answered each of your points below.
1) However, the range of gestational age of the participants is very broad and thus huge differences in the subgroups (if analyzed) are expected. If the authors would stratify the whole group by subgroups according to GA (eg 24 - 26, 27 - 29, 30 - 32 weeks), the value of the research could have been greatly increased. However, the number of children involved should be significantly greater - perhaps this issue could be solved by multicentre study, with other Polish tertiary perinatal centers included. Such design could also shorten the time needed to recruit sufficient number of participants (predicition of > 3 years seems very long)
According your advice we add but two subgroups depending of GA (24-28 and 29-32). Firstly, we decided to do a single center study (unfortunately the quantitive ultrasound equipment is only available in our hospital in Poland and measurement is during hospital stay), but in case of problem of recruiting process we are open for future collaborations with other Polish tertiary perinatal centers. Thank You for suggestion.
2) The authors wrongly define very low birth weight as a weight < 1000 g (see lines 88, 137-138); the correct definition is either extremely low birth weight or change to weight < 1500 g.
We have corrected our mistake, thank You for noticing that.
3) In 2.2 (Setting and participants) I am missing more information about the institution (such as tertiary perinatal center?, annual number of births with GS < 33 weeks?, inborn - outborn / prenatal transport?). Perhaps also including the information on the guidelines regarding maternal supplementation of vitamin D in pregnancy in Poland (if they exist - or a comment that there is no uniform recommendation regarding vit D supplementation in pregnancy).
According your advise we have added the information about our Department and Guidelines of Polish Society of Gynecologists and Obstetricians according to supplementation in pregnancy.
4) As (according to my experience) most ELBW infants need at least 7-10 days to reach 40 mL/kg/day, this limit might mean very late introduction of vitamin D. Perhaps the authors could consider a lower threshold and thus earlier initial dose?
According your advice we decided give oral vitamin D supplementation faster when infant achieve 30ml/kg/day of enteral feeding (minimal enteral feeding). Although the vitamin D supplementation is included in parenteral nutrition from the first days of life (lower dose) and in many centres neonatologists start drops supplementation when they finish parenteral feeding, so even later.
5) In 2.4 I miss criteria for the diagnosis of cholestasis.We have added the criteria for diagnosis of cholestasis.
6)In 2.6 some data on administration of vit D should be added - if oral, is it given directly to the mouth (as I assume from the Devikap product specification, 1 drop = 500 IU) or by gastric tube (directly or diluted in some milk?). As vitamin D binds to the plastic wall this could have a significant effect on the actual amount of the vitamin received by the patient.
We have added data on administration of vitamin D (oral by orogastric tube until the orogastric tube is used or when infants start bottle feeding directly by mouth; just before feeds).
7) In Table 1, I suggest the term Serum concentration of 25(OH)D - instead of vitamin D (in the 1st column). The 2ndcolumn number should probably be <20 ng/mL (or a column 10 - 12 ng/mL is missing?). There are mistakes in the units in the columns with 50-75 and >75 ng - should be mL, not dL.
Thank You for Your suggestion, we have made changes in Table1 and corrected our mistake.
8) The limit of >100 ng/mL (line 152) differs from the highest level in Table 1.
We have improved Table 1 of one column more to better understanding.
9) The 2.7 (Primary outcome) should be combined with Table 1 - as it is mostly repetition of the same data.
We have shortened the definition of primary outcome to avoid repetition.
10) I find citing "neonatal nurses" at each blood collection redundant (lines 189, 224, 234, 254) as the type of sample (venous, capillary, arterial) is important, as well as the quantity of blood (which is not listed!) and not the profession of the person who withdraws it.
We have added quantity of blood, type of sample is also listed and changed "neonatal nurses" to "neonatal staff".
11) The diagnosis for nephrolithiasis (or better - nephrocalcinosis, as stated in line 264) made by "a trained ultrasonographer" might be subjective and biased - I advise that two independent (and blinded for the group and the result of the other ultrasonographer) assessors are included.
We have added two independent ultrasonographers.
12) Table 2 is insufficiently clear and descriptive - particularly as Enrollment and Allocation should be done in the first 24 hours (see line 99), lines Intervention are even blank (or something is missing).
We have made some changes in Table 2 to clarify.
13) As there are new data on analytical unreability of 25-OH-D vitamin measurements in preterm neonates (see Miller JJ et al, J Appl Lab Med 2023) I advise at least mentioning this bias in the discussion - on the other hand, inclusion of the value of ionized calcium as another laboratory parameter in the methods would be advised.
We have mentioned Miller JJ et al in the discussion. We will monitor the calcium serum level measuring Ca-P metabolism.
14) Although an extensive list of relevant literature is provided, I miss some recent citations, such as Jamali Z et al, BMC Pediatr 2023; Kumar M et al, Pediatrics 2022; Malone Jenkins S et al, Pediatr Nephrol 2022; Zung A et al, J Pediatr Endocrinol Metabol 2022.
Thank You for suggesting new literature. We have read mentioned articles and added them to our paper.
15) Reference 16 is incomplete/false - could not be found in PubMed.
Thanks, we have already completed the references (currently 17). Here is the PubMed link to the reference too: https://pubmed.ncbi.nlm.nih.gov/30285373/
Reviewer 2 Report
Comments and Suggestions for Authors
Thanks to the authors for submitting their interesting manuscript entitled “The high initial dose of monitored vitamin D supplementation in preterm infants (HIDVID trial): study protocol for a randomized controlled study”. I have carefully reviewed the manuscript. This is very interesting research. I want to provide them with feedback and recommendations that could enhance the manuscript’s clarity and impact.
Please find my comments below.
· The title needs to be reaffirmed in order to contain the results
· Line 88 and 137: extremely low birth weight is the term for babies< 1000 g
· Table 1: in the line presenting serum concentration of vitamin D, <10ng/mL, 20-30 ng/mL, something seems wrong; maybe the correct is <20ng/mL.
· Line 188: “Neonatal nurses will collect venous samples..” should be changed to “neonatal staff..”
·
I hope these suggestions contribute to refining the manuscript further. Overall, the study is impressive and contributes significantly to the field.
Comments on the Quality of English LanguageMinor editing of English language required
Author Response
Thank you for your review of our paper. We have answered each of your points below.
1) The title needs to be reaffirmed in order to contain the results
We are not sure if we properly understand your suggestion but when we public results our manuscript will be titled as “The high initial dose of monitored vitamin D supplementation in preterm infants: a randomized controlled study”
2) Line 88 and 137: extremely low birth weight is the term for babies< 1000 g
We have corrected our mistake, thank You for noticing that.
3) Table 1: in the line presenting serum concentration of vitamin D, <10ng/mL, 20-30 ng/mL, something seems wrong; maybe the correct is <20ng/mL.
We have changed that, thank You for Your suggestion.
4) Line 188: “Neonatal nurses will collect venous samples..” should be changed to “neonatal staff.”
We have changed "neonatal nurses" to "neonatal staff".
Round 2
Reviewer 1 Report
Comments and Suggestions for Authors
The authors have included all my comments and corrected / supplemented the contents of the paper. I have no further suggestions for the improvement of the contribution.